# MHC Class II Expression Influences the Composition and Distribution of Immune Cells in the Metastatic Colorectal Cancer Microenvironment

**DOI:** 10.3390/cancers14174092

**Published:** 2022-08-24

**Authors:** Brian D. Griffith, Simon Turcotte, Jenny Lazarus, Fatima Lima, Samantha Bell, Lawrence Delrosario, Jake McGue, Santhoshi Krishnan, Morgan D. Oneka, Hari Nathan, J. Joshua Smith, Michael I. D’Angelica, Jinru Shia, Marina Pasca Di Magliano, Arvind Rao, Timothy L. Frankel

**Affiliations:** 1Department of Surgery, University of Michigan, Ann Arbor, MI 48109, USA; 2Department of Surgery, Centre Hospitalier de l’Université de Montréal, Montreal, QC H2X 3E4, Canada; 3Department of Computational Medicine and Bioinformatics, University of Michigan, Ann Arbor, MI 48109, USA; 4Department of Electrical and Computer Engineering, Rice University, Houston, TX 77005, USA; 5Department of Surgery, Memorial Sloan Kettering Cancer Center, New York, NY 10065, USA; 6Department of Pathology, Memorial Sloan Kettering Cancer Center, New York, NY 10065, USA; 7Department of Radiation Oncology, University of Michigan, Ann Arbor, MI 48109, USA; 8Department of Biomedical Engineering, University of Michigan, Ann Arbor, MI 48109, USA; 9Department of Biostatistics, University of Michigan, Ann Arbor, MI 48109, USA

**Keywords:** colon cancer, multiplex immunohistochemistry, immuno-oncology

## Abstract

**Simple Summary:**

Extensive data exist regarding the importance of major histocompatibility complex (MHC) class I in the tumor microenvironment, but data on MHC class II (MHC-II) are lacking. Using multiplex immunohistochemistry and spatial modeling, we demonstrate that MHC-II expression impacts both the relationships of cells traditionally associated with T lymphocyte priming and spatial interactions of cytotoxic lymphocytes and tumor cells in colorectal cancer.

**Abstract:**

Despite advances in therapy over the past decades, metastatic colorectal cancer (mCRC) remains a highly morbid disease. While the impact of MHC-I on immune infiltration in mCRC has been well studied, data on the consequences of MHC-II loss are lacking. Multiplex fluorescent immunohistochemistry (mfIHC) was performed on 149 patients undergoing curative intent resection for mCRC and stratified into high and low human leukocyte antigen isotype DR (HLA-DR) expressing tumors. Intratumoral HLA-DR expression was found in stromal bands, and its expression level was associated with different infiltrating immune cell makeup and distribution. Low HLA-DR expression was associated with increased intercellular distances and decreased population mixing of T helper cells and antigen-presenting cells (APC), suggestive of decreased interactions. This was associated with less co-localization of tumor cells and cytotoxic T lymphocytes (CTLs), which tended to be in a less activated state as determined by Ki67 and granzyme B expression. These findings suggest that low HLA-DR in the tumor microenvironment of mCRC may reflect a state of poor helper T-cell interactions with APCs and CTL-mediated anti-tumor activity. Efforts to restore/enhance MHC-II presentation may be a useful strategy to enhance checkpoint inhibition therapy in the future.

## 1. Introduction

In metastatic colorectal cancer (mCRC), the tumor microenvironment (TME) is composed of a combination of both proinflammatory and immunosuppressive cells, the proportion and distribution of which influences the overall immune state of the tumor [1]. Patients with a greater population of cytotoxic lymphocytes (CTLs) tend to have more favorable outcomes relative to those with a higher proportion of suppressive cells, who succumb to earlier disease recurrence [2,3]. Multiple factors influence the activity of CTLs in the microenvironment. Major histocompatibility complex class I (MHC-I) is one such factor and must be present on the cell surface to allow for recognition of an antigen by the T-cell receptor. Partial or total loss of MHC-I is a common tumor escape mechanism with absence on the cell surface rendering tumor cells less susceptible to immune clearance [4,5,6]. In primary CRC, loss or aberrant MHC-I expression is present in up to 74% of tumors [7,8], and low levels of expression of MHC-I are an independent predictor of poor prognosis [9,10]. In CRC liver metastases and primary tumors, high levels combined with high intratumoral T-cell infiltration are associated with improved survival [10,11]. 

A second and equally important component of immune cell recognition is major histocompatibility complex II (MHC-II), which is expressed on the surface of antigen-presenting cells (APCs) and can be upregulated on nucleated cells, including cancer cells, in response to interferon stimulation. MHC-II serves as the primary cell surface display machinery for phagocytized proteins. MHC-II, including human leukocyte antigen isotype DR (HLA-DR), one of the three heterodimer polypeptide subtypes that compose MHC-II, is essential for effective antigen presentation to CD4^+^ T helper cells, and subsequent priming of naïve CD8^+^ T cells to CTLs. MHC-II expression is also variable in primary CRC [12], with expression in the microenvironment of two-thirds of tumors [13]. In primary CRC, increased MHC-II expression is associated with both improved prognosis [13,14,15,16] and increased tumor-infiltrating lymphocytes [14], and its expression portends a survival benefit when expressed in the colorectal carcinoma epithelium and adjacent non-carcinoma epithelium [16]. While the impact of MHC-I expression on immune infiltration and function in the tumor microenvironment has been studied in CRC liver metastases, there is a paucity of data on MHC-II expression. 

We and others have shown that multiplex fluorescent immunohistochemistry (mfIHC) is used for both phenotype cells and provides a spatial context in the TME, allowing for measurement of cell-to-cell proximity and contact [17,18,19,20,21]. Prior application of mfIHC to CRC demonstrates that increased mixing of tumor cells and CTLs is associated with a proinflammatory TME with a higher engagement of CTLs, as well as enhanced interactions with APCs and T helper cells [18]. Interestingly, tumors with increased mixing of tumor cells and CTLs have immunosuppressive elements, including elevated expression of programmed death receptor ligand (PD-L1^+^)-positive APCs, which is likely a compensatory phenomenon [18]. We sought to use mfIHC to better elucidate the impact of MHC-II expression in the TME of CRC liver metastases in the hopes of identifying novel immune-based approaches to therapy.

## 2. Materials and Methods

### 2.1. Patient Selection

The study population consisted of 195 patients who underwent consecutive curative intent resections of colorectal liver metastasis. Studies were approved by an Institutional Review Board, and patient and tumor characteristics were securely maintained. A gastrointestinal pathologist reviewed whole tissue samples and selected 0.6 mm-diameter cores in triplicate to create a tissue microarray (TMA). Because of tissue folding or core drop-out, 46 tumor samples were not suitable for analysis. As a result, 149 patients were included in the final analysis. Patients who received preoperative chemotherapy received standard treatment as previously described [22]; no patients received immunotherapy, and adjuvant chemotherapy regimens were not routinely recorded.

### 2.2. Histological Analysis

Tissue samples were fixed overnight in neutral-buffered formalin, dehydrated, and embedded in paraffin. Tissue sections were cut at five-microns, deparaffinized, and rehydrated through a gradient of xylene and ethanol baths. Tissues were then stained with hematoxylin and eosin for light microscopic examination.

### 2.3. Multiplexed Fluorescent Immunohistochemistry Staining and Imaging

Five-micron slices were cut from the TMA onto charged slides for processing. Slides were backed at 60 °C for 1 h and underwent deparaffinization and rehydration followed by staining as previously described [17]. In brief, after rehydration, TMA slides were fixed with formalin and subjected to the first round of antigen retrieval buffer with a pH of 9 (AR9, Akoya Biosciences, Marlborough, MA, USA). Multiple rounds of staining were performed and separated by antigen retrieval steps using an antigen retrieval buffer with a pH of 6 and pH of 9 (AR6 and AR9, Akoya Biosciences). Each antigen retrieval was followed by primary and secondary antibody and fluorescent tyranamide signal amplification (TSA, Akoya Biosciences). This allowed for the removal of the prior primary and secondary antibody, while the fluorophore remained covalently bonded to the tissue antigen, ultimately forming a multiplex. The primary antibodies used included CD3, CD8, FoxP3, CD163, PD-L1, pancytokeratin, HLA-DR, granzyme B, and Ki-67 (Opal polymer, Akoya Biosciences) (details on antibodies included in Appendix A). Spectral 4′-6-diamidino-2-phenylindole (DAPI) was used as a counter stain as previously described [17]. Slides were mounted, cover-slipped, and dried overnight. Cores were then imaged using the Mantra Quantitative Pathology Workstation at 20 times magnification with the following channels: DAPI, FITC, CY3, CY5, CY7, Texas Red, and Qdot with an exposure of 250 milliseconds.

### 2.4. Image Analysis: Phenotyping and Cell-To-Cell Interactions

Images were analyzed using the inForm Cell Analysis software (Akoya Biosciences). Images were stratified based on the percentage of cores that expressed HLA-DR into high HLA-DR expression (above the mean, n = 75) or low HLA-DR expression (below the mean, n = 74). Normality of distribution was assessed with a Shapiro–Wilk test. Because the threshold of a biologically important level of expression of HLA-DR is unknown, utilization of the mean avoided potential bias of results. Simple and complex phenotyping was performed, and cell-to-cell interactions, including the distance of a cell to its nearest neighbor and the number of cells engaged with other cells within a set radius, were calculated as previously described [17]. The following phenotypes were assigned: T cell (CD3^+^), antigen-presenting cell (CD163^+^), epithelial cell (EC) (pancytokeratin^+^), and other cells (CD3^−^ CD163^−^ pancytokeratin^−^). Additionally, cytotoxic T cells (CD3^+^ CD8^+^), helper T cells (CD3^+^ CD8^−^ FoxP3^−^), and regulatory T cells (CD3^+^ CD8^−^ FoxP3^+^) were defined.

### 2.5. Image Analysis: Spatial G-Function and Engagement

A G-function was calculated to quantify the spatial relationships and interactions among two or more types of cells in the TME, as previously described [18]. In brief, the G-function was a function of distance and computed the probability of having a non-reference cell type within a certain distance of a reference cell type. It can be mathematically expressed using the following equation:$$G{\left(r\right)}_{x,y}=1-e^{-\lambda_{y}{\pi r}^{2}}$$
where the subscripts ‘x’ and ‘y’ indicate that the spatial distribution of cell type ‘y’ relative to the cell type ‘x’ is being computed, ’r’ refers to the distance from the reference cell type, and $\lambda_{y}$ the overall density of cell type ‘y’ on the slide. To correct for edge effects, Kaplan–Meier correction is applied to the computed G-function. The area under the curve (AUC) metric was used to characterize the rate at which the G-function rose, which was shown to be prognostic of outcomes in non-small cell lung cancer [19] and intraductal mucinous neoplasms [20]. The AUC was calculated using a radius of 60 microns.

### 2.6. Statistical Analysis

Statistical analyses were performed with JMP Pro 13.2.0 unless otherwise stated. Differences in cell phenotype, intercellular distance, cell engagement, or AUC were evaluated by a two-sided analysis of variance (ANOVA). Data that were not normally distributed were evaluated by non-parametric Wilcoxon rank-sum. Categorical variables were analyzed with Fisher’s exact test. *p* ≤ 0.05 was considered significant and was adjusted for multiple testing using the Benjamini–Hochberg false discovery rate procedure when necessary. For survival analysis, Kaplan–Meier plots were drawn, and statistical differences were determined by log rank. 

## 3. Results

### 3.1. High HLA-DR Expression Was Associated with Decreased Distance between T Helper Cells and Antigen-Presenting Cells

To study the effect of MHC-II expression on the immune microenvironment in mCRC, IHC for HLA-DR was performed on 149 colorectal liver metastases from patients who underwent curative intent resection (Figure 1A). Patients were stratified into two groups: high expression of HLA-DR (above the mean, n = 75) and low expression of HLA-DR (below the mean, n = 74). No significant difference was found with respect to sex, age, mean tumor size, tumor number, disease-free interval, clinical risk score, N stage, extra-hepatic disease, or use of preoperative chemotherapy between the two groups (Table 1). Kaplan–Meier analysis revealed no significant difference in overall survival (Appendix A). Assessment of microsatellite instability (MSI) was performed by IHC for MLH1, MSH2, MSH6, and PMS2. Patients that were deemed MSI high had significantly greater surface area positive for HLA-DR (Appendix A). Samples were then subjected to mfIHC for the markers CD3, CD8, FoxP3, CD163, pancytokeratin, and PD-L1. InForm software (Akoya Bioscience) analysis was used to phenotype immune and tumor cells in the microenvironment and assign each cell a unique spatial location. These data were then used to calculate intercellular distances. A representative example of an H&E, mfIHC composite imaging, cellular phenotyping, and measurement of intercellular distance is shown in Figure 1, with representative examples of high HLA-DR expression in Figure 1A and low HLA-DR expression in Figure 1B. To determine the impact of HLA-DR expression on the spatial relationships of T cells and APCs, intercellular distances were reported. Patients with high HLA-DR expression demonstrated a lower mean distance from T cells to the nearest APC with a mean distance of 46.01 μM compared to 61.65 μM in the low expressing cohort (*p* = 0.0033, Figure 1C). More specifically, there is a shorter mean distance from helper T cells to APCs (50.43 μM in HLA-DR high tumors vs. 66.01 μM in HLA-DR low tumors, *p* = 0.0039, Figure 1D). When stratified by the surface area of expression, there was a direct correlation between HLA-DR-positive surface area and the intercellular distances of APCs to T cells (R = 0.1381, *p* < 0.0001, Figure 1E) and helper T cells (R = 0.1156, *p* < 0.0001, Figure 1F). 

### 3.2. High HLA-DR Expression Was Associated with Greater Engagement between T Cells and APCs

While intercellular distance describes the overall spatial relationships between cells, cellular engagement better represents the physical interaction between cells and, therefore, receptor/MHC contact [17]. To determine cellular engagement, we identified circumstances in the TME in which a T cell was within 40 μm of the center of an APC. Representative images of an H&E, mfIHC composite image, cellular phenotype, and map of intercellular engagement are shown in Figure 2. Representative maps of intercellular engagement showed increased overlap of APCs and helper T cells in a tumor with high HLA-DR expression (Figure 2A) relative to one with low HLA-DR expression (Figure 2B). Calculating intercellular engagement across the entire cohort, we found increased engagement of APCs and T cells (64.34% of T cells in HLA-DR high tumors vs. 50.66% of T cells in HLA-DR low tumors, *p* = 0.0004, Figure 2C) and more specifically engagement of helper T cells and APCs (59.89% of helper T cells in HLA-DR high tumors vs. 47.51% of helper T cells in HLA-DR low tumors, *p* = 0.0007, Figure 2D). Similarly, the percent of APCs engaged with helper T cells positively correlated with the percent positive area of HLA-DR expression (R = 0.1752, *p* < 0.0001, Figure 2E). When looking at each T helper cell that was engaged, cells tended to be in contact with a greater number of APCs simultaneously when HLA-DR expression was high (R = 0.1628, *p* < 0.0001, Figure 2F). Taken together, these findings suggest that increased HLA-DR expression is associated with decreased distance and increased engagement between T cells, specifically helper T cells, and APCs. 

### 3.3. HLA-DR Expression Is Associated with Changes in the Spatial Relationship of Cells Not Associated with Traditional MHC-II Interactions

MHC-II expression is essential for the effective presentation of antigens to T helper cells, which are subsequently integral in the priming of naïve CD8^+^ T cells into CTLs. To determine the impact of HLA-DR expression on the interaction of tumor epithelial cells (ECs) and T cells in the TME, the intercellular distance and engagement among T cells and ECs were described. HLA-DR expression was associated with a lower mean distance between all T cells and ECs (75.30 μM in high HLA-DR tumors vs. 93.04 μM in low HLA-DR tumors, *p* = 0.0015, Figure 3A). Similarly, the percent of ECs engaged with T cells positively correlated with the percent expression of HLA-DR (R = 0.3473, *p* < 0.0001, Figure 3B). 

As engagement of CTLs with tumor ECs was associated with improved survival [17], we set to specifically determine the impact of HLA-DR expression on the interaction of these two cell phenotypes. This was of particular interest as CTL surface receptors typically interact with MHC-I and not MHC-II antigen-presenting machinery. HLA-DR expression was associated with improved interaction among CTLs and tumor ECs, as evidenced by both lower mean intercellular distance (156.50 μM in high HLA-DR tumors vs. 220.37 μM in low HLA-DR tumors, *p* < 0.0001, Figure 3C) and increased cellular engagement (23.19% and 14.19% of CTLs in high HLA-DR and low HLA-DR, respectively, *p* = 0.0003, Figure 3D). To assess the contribution of APC/helper T-cell interaction to this finding, a bivariate analysis was performed and revealed a strong correlation between their engagement and that of tumor cells and CTLs (Figure 3E). We next looked at the association of HLA-DR expression on T regulatory cells (Treg). Paradoxically, tumors with high HLA-DR expression showed greater engagement between Tregs and CTLs and less intercellular distance (Appendix A). 

Programmed death receptor-1 (PD-1) and its ligand (PD-L1) inhibit the response of T cells and impact their spatial relationships with tumor cells. To determine if interactions were even more pronounced in tumors lacking PD-L1, cells were stratified by expression. CTL engagement with PD-L1^−^ ECs was increased in tumors with high HLA-DR expression (18.39% of CTLs in high HLA-DR tumors vs. 6.49% in low HLA-DR tumors, *p* = 0.0001, Figure 3F), but this effect was lost with PD-L1^+^ ECs where there was no difference in CTL engagement by HLA-DR expression (*p* = 0.7589, Figure 3G). These data suggest that MHC-II expression plays an integral role in not only the classical interactions of APCs and helper T cells but the engagement of CTLs and tumor ECs in the TME of mCRC. The abrogation of this finding with PD-L1^+^ ECs supports the hypothesis that HLA-DR may prime traditional CTLs for more effective function. 

In addition to influencing CTLs interactions with tumor ECs, there was a significantly higher interaction between CTLs and APCs in the TME of tumors with high HLA-DR expression (32.8% of CTLs in high HLA-DR tumors vs. 14.3% in low HLA-DR tumors, *p* < 0.0001, Figure 3H).

### 3.4. HLA-DR Expression Is Associated with Greater Infiltration of Immune Cells in the TME 

The infiltration of both CTLs [23] and T helper cells [24] in the TME favor local immune activation and are thought to correlate positively with the outcome. We, therefore, assessed the influence of HLA-DR expression on the infiltration of immune cells. HLA-DR expression was associated with an increased proportion of immune cells among all cells in the TME (17.06% in high HLA-DR tumors vs. 7.78% in low HLA-DR tumors, *p* < 0.0001, Figure 4A). This relationship held true when specifically assessing the percentage of T cells (6.6% vs. 2.67% in high HLA-DR and low HLA-DR tumors, respectively, *p* < 0.0001, Figure 4E), as well as T helper cells (4.16% in high HLA-DR tumors vs. 2.21% in low HLA-DR tumors, *p* = 0.0007, Figure 4B) and CTLs (1.6% in high HLA-DR tumors vs. 0.19% in low HLA-DR tumors, *p* = 0.0002, Figure 4F). Stated another way, when stratified by the degree of T helper cell infiltration, tumors with high infiltration of T helper cells were associated with an increased degree of HLA-DR surface staining (28.12% in high infiltration vs. 10.43% in low infiltration tumors, *p* = 0.0003, Figure 4C). The percentage of T helper cells of all cells in the TME was positively correlated with the degree of HLA-DR staining (R = 0.1426, *p* < 0.0001, Figure 4D). Additionally, the ratio of CTLs to Tregs was increased in tumors with high HLA-DR expression (24.9 vs. 6.2 in low HLA-DR tumors, *p* = 0.0234, Figure 4G). These data may suggest that HLA-DR expression is integral to the infiltration of immune cells, including both T helper cells and CTLs, in the microenvironment of mCRC. However, it should be noted that immune cells, particularly APCs and activated T helper cells, express HLA-DR, and, therefore, these observations may simply relate to the infiltration of a greater number of HLA-DR expressing cells. 

### 3.5. HLA-DR Expression Level Is Associated with Distinct Cell Population Mixing in the TME

G-function is established as a method to describe the population-level mixing of two or more cell types in the TME [18,19,20]. The rate of rising of the G-function can be used as a surrogate to measure the degree of cellular mixing, and the area under the curve (AUC) metric can therefore be used to compare differences in cellular mixing at a fixed radius from individual cells, where a high AUC correlates to a high degree of mixing [18]. Figure 5 shows a representative cellular phenotype map of a high HLA-DR expressing tumor (Figure 5A) with abundant mixing of helper T cells and APCs and a low HLA-DR expressing tumor with little mixing (Figure 5B). After calculating G-function curves for these two cell types, we found significantly greater population mixing in high HLA-DR expressing tumors (G-function AUC = 1.21 vs. 0.36 in low HLA-DR tumors, *p* = 0.0013, Figure 5C). Additionally, representative images of high (Figure 5D) and low (Figure 5E) HLA-DR tumors demonstrated that high HLA-DR expression was associated with increased cellular mixing among tumor ECs and CTLs (G-function AUC = 2.71 in high HLA-DR tumors vs. 0.63 in low HLA-DR tumors, *p* = 0.0004, Figure 5F). 

### 3.6. HLA-DR Expression Influences the Proportion of Activated Immune Cells in the TME

Ki67 is a nuclear protein that is used to identify cellular proliferation and can serve as a marker of T-cell activation. Similarly, staining for granzyme B (GZMB), one of the functional elements of cytotoxic CD8^+^ T cells, can be utilized to identify activated CTLs. To assess the influence of HLA-DR on T helper and CTL activity, mfIHC was performed on the tumor cohort for CD3, CD8, Ki67, GZMB, and pancytokeratin (representative image, Figure 6A). Tumors with high HLA-DR expression had a similar proportion of activated T cells (GZMB^+^Ki67^+^) with 3.52% activated compared to 2.26% in HLA-DR low tumors (*p* = 0.7673, Appendix A). Interestingly, there was an increased engagement of tumor ECs with activated CTLs (19.31% in high HLA-DR tumors vs. 9.84% in low HLA-DR tumors, *p* = 0.0039, Figure 6B). The proportion of activated T helper cells was also increased with HLA-DR expression (5.78% vs. 3.76% in high HLA-DR vs. low HLA-DR tumors, respectively, *p* = 0.0364, Figure 6C), and the proportion of activated T helper cells positively correlated with the degree of HLA-DR staining (R = 0.0388, *p* = 0.0188, Figure 6D). Additionally, HLA-DR expression was associated with an increase in the engagement of APCs to activated T helper cells (9.11% vs. 3.90% of activated helper T cells in high HLA-DR vs. low HLA-DR tumors, respectively, *p* = 0.0137, Figure 6E). To determine the impact of APC/helper T-cell engagement on CTL function, patients were dichotomized around the median into high and low. Tumors with high engagement had significantly greater CTL activity, as evidenced by increased engagement of epithelial cells to activated CTLs (21.3% in high APC/helper T-cell engagement vs 9.3% in the low group, *p* = 0.0002, Figure 6F). These data suggested that MHC-II expression was associated with increased infiltration and proportion of activated immune cells in the TME of mCRC.

## 4. Discussion

Metastatic colorectal cancer remains a highly morbid and fatal disease that is rarely cured, despite significant advances in therapy over the past decades. Immune-based therapies have been effective only for a small subset of patients with mCRC [25], highlighting the need to better understand the immune microenvironment. Prior studies of the immune microenvironment have primarily investigated only the predominant cell type and proportion of infiltration, establishing that greater numbers of proinflammatory cells greatly impact survival [2,26]. Greater infiltration of CTLs is associated with both anti-tumor activity [27] and a favorable prognosis [21]. As is true for most tumor types, effective CTL infiltration and function depends on many factors, including local cytokines/chemokines, presence of surrounding suppressive elements, and proper processing and display of peptides by both tumor cells and professional antigen-presenting cells such as macrophages and dendritic cells. Expression of the primary presenting molecules for CTLs via MHC-I is necessary for activation of CTLs, and partial or total loss of MHC-I is thought to be a common tumor escape mechanism [4]. The impact of MHC-I expression on the cellular makeup of the TME in mCRC has been studied with a low level of expression portending a poor prognosis, while high levels of expression are associated with greater T-cell infiltration and overall survival [11]. While MHC-I is primarily responsible for recognition and activation by CTLs, MHC-II molecules, displaying peptides from phagocytized extracellular proteins, bind to cognate receptors on helper T cells resulting in cytokine release, which primes CTL function. Although its expression is known to be variable in CRC [12] and increased expression is associated with an improved overall prognosis [13,14,15], there is a paucity of data on the impact of MHC-II expression on the immune composition of the CRC TME. 

Much of the prior work evaluating the TME in mCRC utilized either flow cytometry or standard IHC to describe the relative abundance or scarcity of cell types [28,29,30]. The former allows for robust multiantigen cell phenotyping, but reliance on single cell suspension ignores key spatial data, which can be crucial for context. IHC captures cell positioning but suffers from an inability to distinguish co-localized antigens making more complex cellular phenotyping impossible. We and others have shown that mfIHC provides the ability to perform multiantigen phenotyping of cells while preserving important spatial data to evaluate the context in which immune cells interact in the TME [17,18,19,20]. Using mfIHC, we have previously demonstrated that engagement and population-level mixing of CTLs with tumor ECs in mCRC is associated with both immune activation and improved disease-specific outcomes [17,18].

In this study, we sought to better define the role of MHC-II in shaping the immune TME by performing mfIHC on a large cohort of patients undergoing curative intent resection for colorectal liver metastases. By utilizing a patient subset that had all macroscopic disease removed, each individual began with the same disease burden eliminating an important confounder for disease-specific outcomes. Tumors were dichotomized around the mean expression of HLA-DR, an isotype of MHC-II, as has been previously performed for MHC-I [11]. Multiantigen phenotyping and analysis of spatial relationships allowed us to explore two important phenomena in the TME, the role of MHC-II expression in APC/helper T-cell interaction and the subsequent priming of CTLs to better recognize tumor cells. 

Multiple prior studies have investigated the association between MHC-II expression and survival. Sconocchia et al. studied 1000 primary CRC tumor samples and identified that 23% of patients with positive HLA-DR staining had favorable disease-specific outcomes [31]. Similarly, Dunne et al. found that patients with HLA-DR-positive tumors had double the survival of those lacking expression [16]. Interestingly, they also found that expression declined with increasing primary cancer stage, which may explain the relatively low level of expression we found in the metastases of patients in our cohort. In our samples, the mean surface area expression of HLA-DR was only 9%, with a range of 0.07–70.92%. Contrary to prior studies, we found no relationship between HLA-DR expression and survival. We hypothesize two potential explanations for this: (1) the microenvironment of metastatic tumors and the role of various cell types differ significantly between metastases and primary tumors (highlighted by the relatively low level of HLA-DR expression in our cohort) or (2) HLA-DR expression in primary tumors may impact metastatic ability, which drives differences in survival seen in the previous studies. The fact that prior reports described decreased HLA-DR expression in later stages supports both hypotheses. 

While these prior studies focused on prognosis, there are limited data on the associations of MHC-II expression and the cellular makeup and interactions in the TME. In a study of 76 primary colon cancers, expression was associated with increased T-cell infiltration, but this included tumors presenting at various stages, which may confound results [32]. Similarly, we found that tumors with higher MHC-II expression had greater infiltration of T cells, including both helper T cells and CTLs. We further demonstrated that in high-expressing tumors, the helper T cells were more engaged with professional APCs, suggesting that intact peptide display machinery may aid in antigen recognition. 

One of the key functions of helper T cells is the production of a cytokine milieu that primes CTLs for activation and clearance of abnormal cells. Phenotyping multiple cell types within the same microenvironment allow the assessment of the impact of one set of cellular interactions on another. This enabled us to demonstrate a strong association between APC/helper T-cell engagement and the activity of CTLs. Co-staining for CD3, CD8, and the intracellular activation markers GZMB and Ki67 revealed the importance of intact MHC-II in the TME on CTL effector function. While these relationships are well known biologically, these are the first data demonstrating that impairment of MHC-II expression impacts the association of putative MHC-I expressing cells and CTLs. This could have important implications for future designs of immunotherapeutics as strategies to increase MHC-II expression and, therefore, helper T-cell activation may improve the efficacy of therapies aimed to support CTL function [33]. Indeed, experimental overexpression of the MHC-II gene promoter (CIITA) in tumor cells results in robust immune activation and tumor rejection [34]. Depletion of helper T cells in these studies abrogated tumor rejection highlighting the importance of MHC-II antigen processing and presentation in tumor immunity. Clinically, interferon alpha, a powerful inducer of both MHC-I and II expression, was used for decades to treat melanoma. Similarly, vaccines designed to be expressed by MHC-II molecules have been effective in mediating tumor clearance in some murine models of cancer [35]. Combinations of therapies aimed at increasing activation of both helper T cells and CTLs through enhancement of MHC-II expression and checkpoint blockade inhibition, respectively, could represent a promising strategy for this difficult-to-treat disease. 

While greater HLA-DR expression was generally associated with a proinflammatory environment, there was a paradoxical increase in regulatory T-cell engagement with CTLs. One potential explanation for this is a compensatory increase in immune suppression that often coincides with inflammation. We previously reported a similar pattern with increased suppressive PD-L1^+^ APC infiltration in metastatic tumors with high CTL activity [17]. The impact of increased regulatory T cells on the TME and potentially survival warrants further investigation. Of note, the ratio of CTLs to regulatory T cells was higher in HLA-DR high tumors, supporting a more proinflammatory microenvironment exists with increased HLA-DR expression.

While this article examined the role of MHC-II expression on T-cell engagement and function with APCs and tumor cells, emerging evidence suggests that B cells and tertiary lymphoid structures (TLSs) also play a role in shaping the TME in mCRC. B cells, and regulatory B cells, in particular, have been shown to influence the inflammatory response in cancer through the secretion of anti-inflammatory mediators [36]. Antigen presentation by B cells via MHC-II is required for optimal effector T-cell function [37,38], and MHC-II molecules appear to be integral for B cell interaction with activated T cells [39]. Additionally, the role of TLSs, aggregates of immune cells in non-lymphoid tissues at sites of chronic inflammation, in shaping the TME in cancer [40] and colorectal cancer specifically [41] is an area of active investigation. Interestingly, TLSs surrounding tumors with increased densities of T helper cells were associated with relapse of advanced colorectal cancer [42]. Thus, the role of MHC-II expression on B cells and in TLS function in the TME of mCRC requires further investigation.

Another unexpected finding from our study was an association between HLA-DR expression and MSI status of tumors. While HLA-DR high tumors were not more likely to be MSI high, there was a greater surface area positivity in this cohort. This finding is not due simply to increased APC infiltration, as we have previously demonstrated a lack of association with MSI status [17]. A possible explanation is that the CTL infiltration associated with MSI-high tumors leads to increased interferon in the TME and subsequent upregulation of MHC-II. Further in vitro and in vivo study is needed to precisely understand this and other associations that have been identified. 

An important limitation of our study is the inability to precisely determine the cellular source of HLA-DR expression in our samples. Both tumor and immune cells are known to express this on the surface, and biological implications of differential expression or loss in either cell type are unknown and require further investigation. Additionally, mfIHC is limited by the ability to query six antibodies per tissue section, which limits the number of cells that can be phenotyped. This prevents the investigation of super-specialized cells, including polarized dendritic cells and B lymphocytes.

## 5. Conclusions

Spatial characterization of immune cells in CRC liver metastases resected in a large cohort of patients demonstrated that intra-tumoral HLA-DR expression was associated with distinct makeup and distribution of immune cells in the tumor microenvironment. High expression of HLA-DR was associated with increased immune cell infiltration and increased proximity between T cells, APCs, and cancer cells. Further work aiming to increase MHC-II expression may be valuable to enhance the effectiveness of currently available CTL-based therapies. 

## Figures and Tables

**Figure 1 cancers-14-04092-f001:**
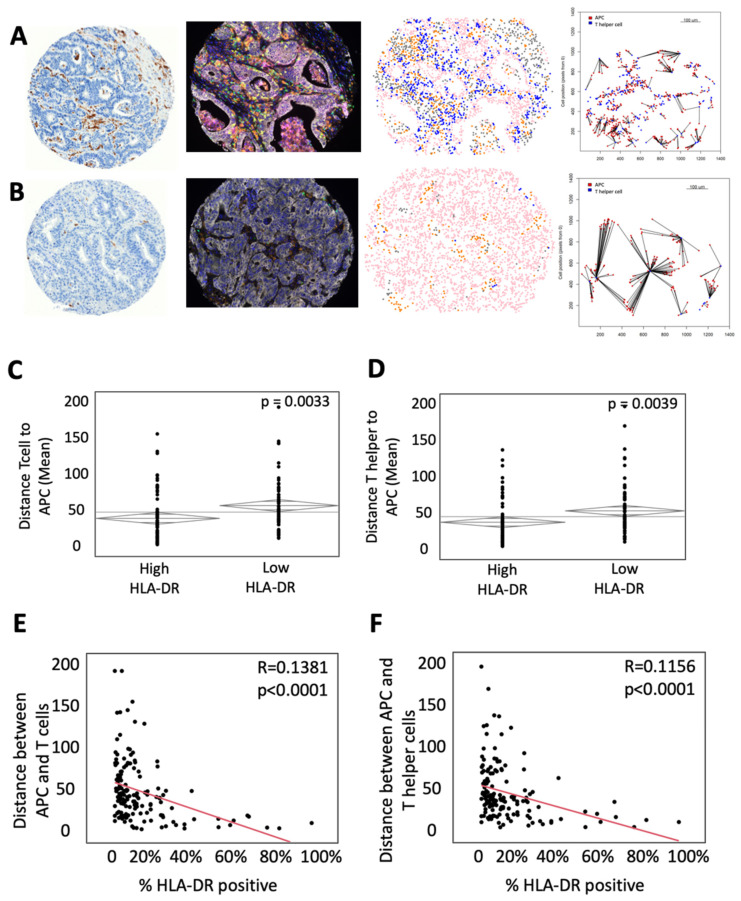
High HLA-DR expression is associated with decreased distance between T helper cells and antigen-presenting cells (APCs). (**A**) Representative example of a tumor with high HLA-DR expression (Left to right: (a) immunohistochemistry for HLA-DR, (b) composite multiplex immunohistochemistry image stained for CD3—green, CD163—orange, CD8—yellow, pancytokeratin—white, FoxP3—red, PD-L1—magenta, and DAPI—blue, (c) phenotypic map depicting the location of CD3^+^ T cells (blue), pancytokeratin-positive epithelial cells (pink), CD163^+^ APCs (orange) and other cells (gray), (d) nearest neighbor analysis between APCs (red) and T helper cells (blue). (**B**) Representative example of a tumor with low HLA-DR expression with images as described above. ANOVA analysis of the mean intercellular distance between (**C**) APCs and T cells and (**D**) APCs and T helper cells. Bivariate comparison of percentage of HLA-DR positivity by surface area and intercellular distances between (**E**) APCs and T cells and (**F**) APCs and T helper cells.

**Figure 2 cancers-14-04092-f002:**
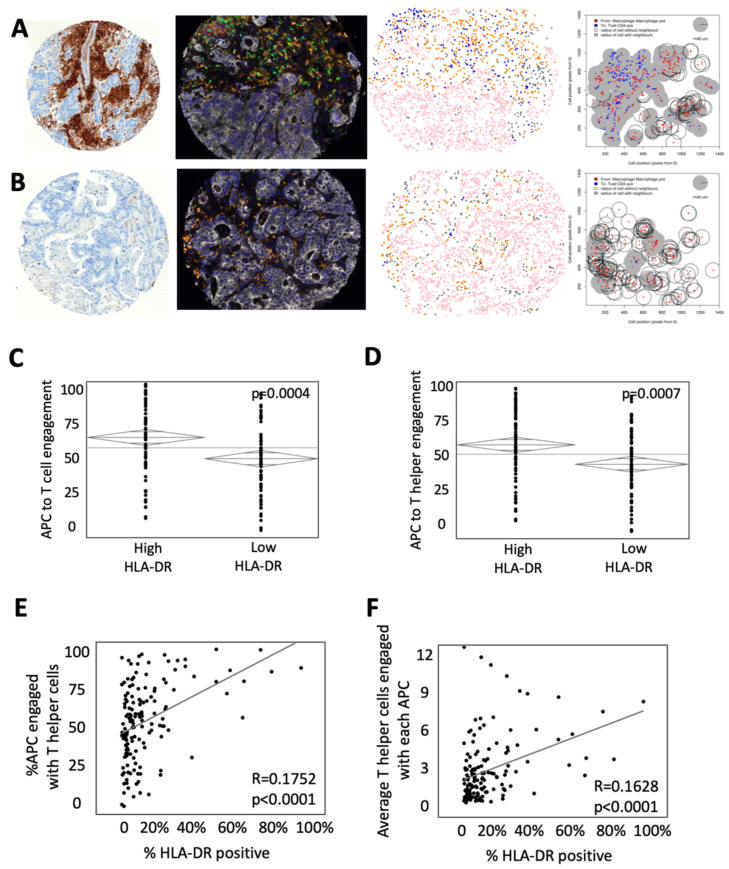
High HLA-DR expression is associated with increased engagement of T cells and antigen present cells (APCs). (**A**) Representative example of a tumor with high HLA-DR expression (Left to right: (a) immunohistochemistry for HLA-DR, (b) composite multiplex immunohistochemistry image stained for CD3—green, CD163—orange, CD8—yellow, pancytokeratin—white, FoxP3—red, PD-L1—magenta and DAPI—blue, (c) phenotypic map depicting the location of CD3^+^ T cells (blue), pancytokeratin-positive epithelial cells (pink), CD163^+^ antigen-presenting cells (orange) and other cells (gray), (d) cell engagement analysis between antigen-presenting cells (red) and T helper cells (blue) with the shaded area representing engaged cells.) (**B**) Representative example of a tumor with low HLA-DR expression with images as described above. ANOVA analysis of the percent of APCs engaged with (**C**) T cells and (**D**) T helper cells. Bivariate comparison of the percentage of HLA-DR positivity by surface area and engagement with (**E**) T cells and (**F**) T helper cells.

**Figure 3 cancers-14-04092-f003:**
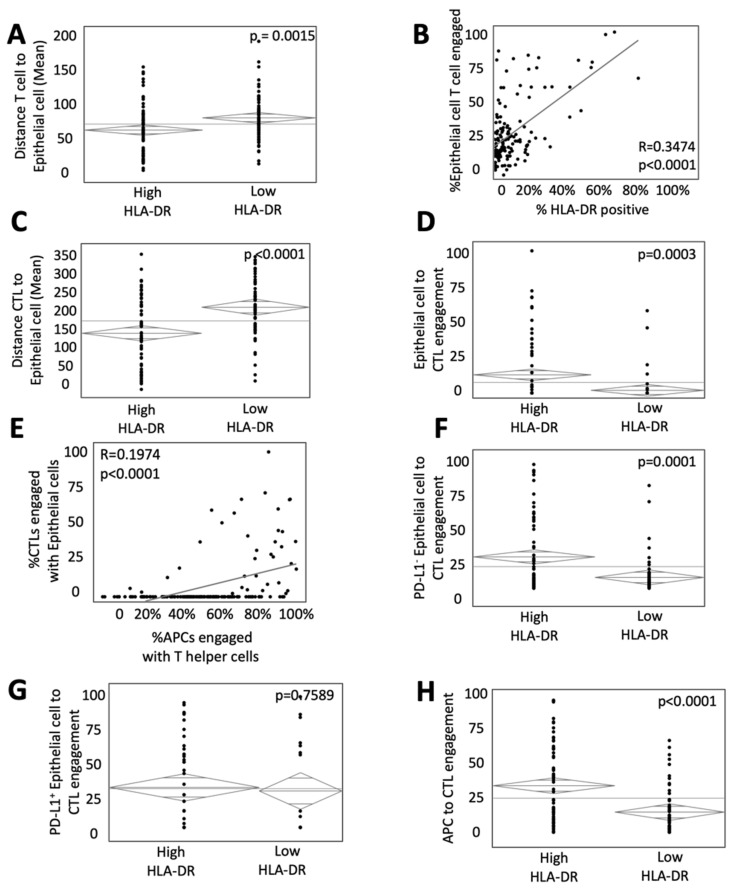
HLA-DR expression influences the spatial relationship of immune and epithelial cells (ECs). (**A**) ANOVA analysis of the intercellular distance between ECs and T cells in high and low HLA-DR expressing tumors. (**B**) Bivariate comparison of HLA-DR-positive surface area and engagement between EC and T cells. ANOVA analysis of (**C**) intercellular distance and (**D**) engagement between T helper cells and ECs. ANOVA analysis of (**E**) intercellular distance and (**F**) engagement between cytotoxic T cells (CTL) and ECs relative to HLA-DR expression. (**G**) Bivariate analysis comparing engagement of CTLs and ECs with APCs to T helper cells. ANOVA analysis of engagement between CTLs and (**H**) PD-L1^−^ and (**I**) PD-L1^+^ relative to HLA-DR expression.

**Figure 4 cancers-14-04092-f004:**
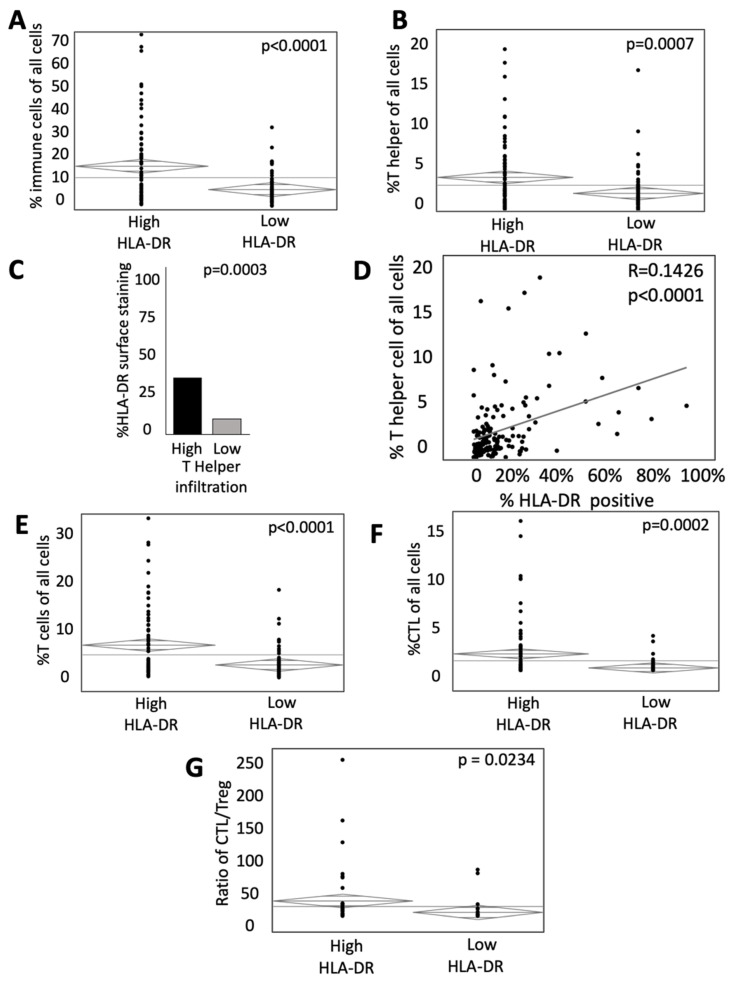
HLA-DR expression influences the proportion of activated immune cells in the TME. (**A**) ANOVA analysis of the percent of immune cells (CD3^+^ T cells and CD163^+^ APCs) relative to all cells in high and low HLA-DR expressing tumors. (**B**) ANOVA analysis of the percent of T helper cells relative to all cells in high and low HLA-DR expressing tumors. (**C**) Percent surface area of HLA-DR staining in patients with high (upper quartile) and low (lowest quartile) T helper cell infiltration. (**D**) Bivariate analysis of HLA-DR-positive surface area and percentage of T helper cells of all cells. Relative abundance of (**E**) total T cells and (**F**) cytotoxic T cells in high and low HLA-DR-expressing tumors.

**Figure 5 cancers-14-04092-f005:**
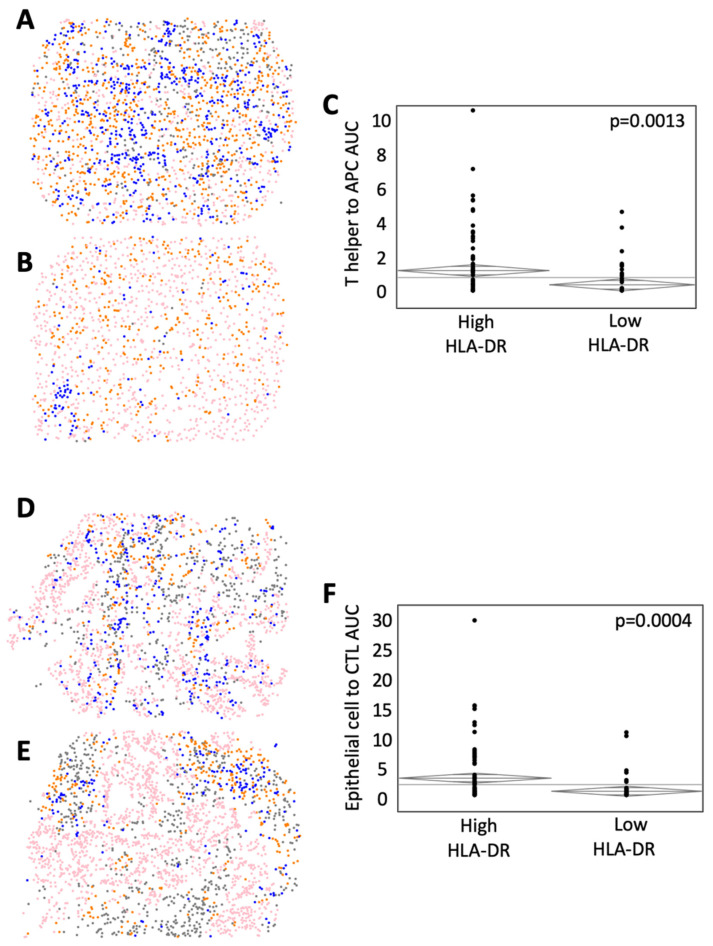
HLA-DR expression is associated with greater immune and epithelial cell population mixing in the TME. Representative phenotypic maps are shown of (**A**) high and (**B**) low T helper cell macrophage mixing (CD3^+^ T cells (blue), pancytokeratin-positive epithelial cells (pink), CD163^+^ antigen-presenting cells (orange), and other cells (gray). (**C**) ANOVA analysis of the area under the curve (AUC) of G-function curves, a mathematical representation of population-level cellular mixing between high and low HLA-DR expressing tumors. Representative phenotypic maps are shown of (**D**) high and (**E**) low CTL to epithelial cell mixing. (**F**) ANOVA analysis comparing G-function AUCs between high and low HLA-DR-expressing tumors.

**Figure 6 cancers-14-04092-f006:**
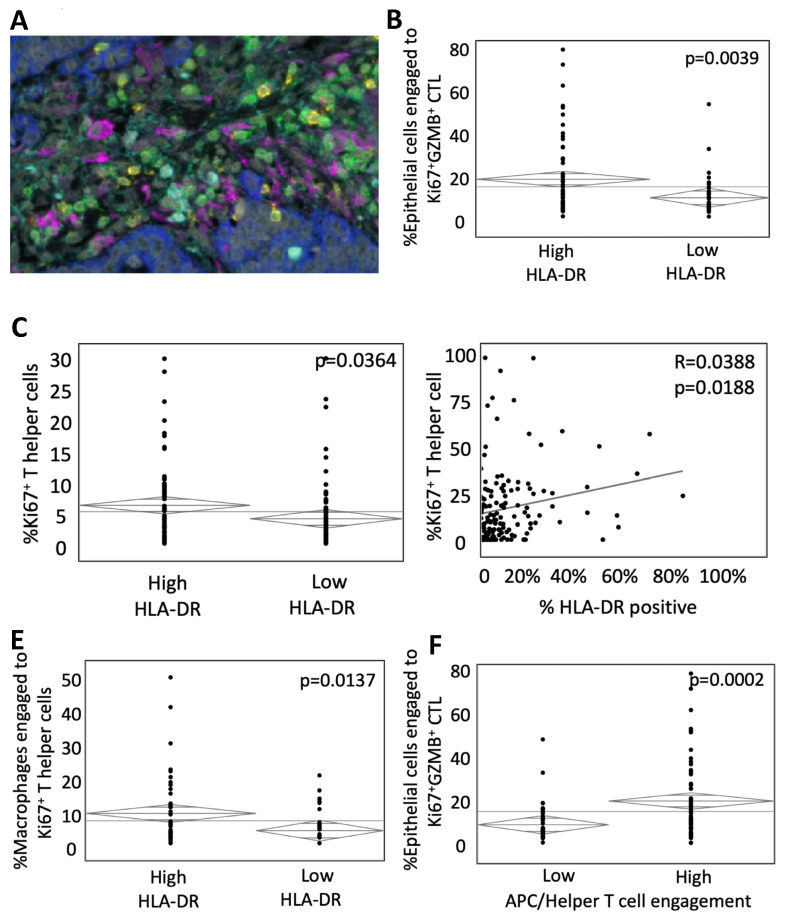
HLA-DR expression influences the proportion and engagement of activated immune cells in the TME. (**A**) Representative multiplex immunohistochemistry image of the metastatic colorectal tumor microenvironment (CD3—green, CD163—magenta, CD8—yellow, pancytokeratin—blue, granzyme B (GZMB)—cyan, and DAPI—white). (**B**) ANOVA analysis of epithelial cells (EC) engaged with activated cytotoxic lymphocytes (Ki67^+^GZMB^+^CD3^+^CD8^+^) in tumors with high and low HLA-DR expression. (**C**) Relative abundance of activated (Ki67^+^) T helper cells. (**D**) Bivariate analysis of activated T helper cells and percent HLA-DR-positive surface area in tumor cores. (**E**) ANOVA analysis of macrophage engagement with activated T helper cells in high and low HLA-DR expressing tumors. (**F**) ANOVA analysis of EC and activated CTL engagement in tumors with high and low engagement of antigen-presenting and T helper cells.

**Table 1 cancers-14-04092-t001:** Demographic data on patients with high and low HLA-DR-expressing tumors.

** Demographic **	** High HLA-DR (n = 74) **	** Low HLA-DR (n = 75) **	** * p * ** ** -Value **
**Gender (%M/F)**	61/39	54/46	0.4072
**Age (mean years)**	58.9	62.2	0.1151
**Tumor size (mean)**	4.7	4.7	0.7371
**Number (<3/≥3)**	73/27	68/32	0.5868
**DFI (mean mos)**	19.1	16.7	0.4166
**CRS** **1** **2** **3** **4** **5**	36 39 19 5 1	36 37 14 13 0	0.3981
**N Stage** **0** **1** **2**	39 38 23	29 47 24	0.3750
**Pre-op chemo**	68%	65%	0.7305
**Extra-hepatic metastases**	16%	10%	0.3278
**MSI high**	8%	4%	0.1481

## Data Availability

Data published in this report are available by request to the corresponding author.

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
