# Peer review of "MHC Class II Expression Influences the Composition and Distribution of Immune Cells in the Metastatic Colorectal Cancer Microenvironment"

_cancers, 2022, doi:10.3390/cancers14174092_

Round 1

Reviewer 1 Report

In their manuscript, Griffith et al. segregate human colorectal liver metastasis samples based on HLA-DR high and low expression. Further using multiplex fluorescent immunohistochemistry, the authors demonstrate enhanced T cells infiltration and engagement of APC-CD4+ T cells in the TME. In addition, the authors show significantly greater engagement between CTL-EC, and Tregs-CTLs in high HLA-DR tumors. The authors also identify enhanced CTL activity in tumors with enhanced APC-CD4+ T cell engagement. 

MHC-II or HLA-DR is a double-edged sword and can contribute to both pro and anti-inflammatory responses. In addition, it has been shown in several studies that the ratio of cytotoxic and suppressor T cells provides best details about the TME. However, in this work the authors show a positive impact a high HLA-DR expression on Ki67 and Granzyme B expression in CTLs. The overall conclusions reached by the authors are supported by the data presented. However, there are few concerns that needs to be addressed. 

Major Points

1-    The authors did not see a correlation between HLA-DR expression and survival in this study and provided possible valuable explanations in the discussion. The reviewer would like to know the ratio of CTL/Tregs in HLA-DR high and low samples 

2-    The authors should also discuss about the possible contribution of B lymphocytes particularly Bregs and Tertiary lymphoid structures in this study. 

3-    The authors observed enhanced EC-CTL engagement in samples with high APC-CD4+ T interactions. The reviewer would like to know the correlation of HLA-DR expression and DC1-CTL engagement. 

4-    Optional: The authors should also study the IL-10 and TGF-beta expression levels in HLA-DR high and low samples.

Minor Points:

1-    The authors did not mention the HLA-DR antibody used for the study.

2-    Inconsistent use of MHC-II & MHC2 and Figure & figure

Author Response

In their manuscript, Griffith et al. segregate human colorectal liver metastasis samples based on HLA-DR high and low expression. Further using multiplex fluorescent immunohistochemistry, the authors demonstrate enhanced T cells infiltration and engagement of APC-CD4+ T cells in the TME. In addition, the authors show significantly greater engagement between CTL-EC, and Tregs-CTLs in high HLA-DR tumors. The authors also identify enhanced CTL activity in tumors with enhanced APC-CD4+ T cell engagement. 

MHC-II or HLA-DR is a double-edged sword and can contribute to both pro and anti-inflammatory responses. In addition, it has been shown in several studies that the ratio of cytotoxic and suppressor T cells provides best details about the TME. However, in this work the authors show a positive impact a high HLA-DR expression on Ki67 and Granzyme B expression in CTLs. The overall conclusions reached by the authors are supported by the data presented. However, there are few concerns that needs to be addressed.

- We thank the reviewer for their thoughtful evaluation of our work and agree it will add to knowledge in the field. We have addressed the concerns below and edited the manuscript to add further information.

Major Points

  • The authors did not see a correlation between HLA-DR expression and survival in this study and provided possible valuable explanations in the discussion. The reviewer would like to know the ratio of CTL/Tregsin HLA-DR high and low samples 

Thank you for this thoughtful suggestion. We have included data demonstrating that HLA-DR high tumors have a higher CTL/Treg ratio (Figure 4G).

  • The authors should also discuss about the possible contribution of B lymphocytes particularly Bregsand Tertiary lymphoid structures in this study. 

Thank you for this suggestion. We have added a paragraph to the discussion on the potential role of Bregs and Tertiary lymphoid structures.

  • The authors observed enhanced EC-CTL engagement in samples with high APC-CD4+T interactions. The reviewer would like to know the correlation of HLA-DR expression and DC1-CTL engagement. 

We thank the reviewer for this important point. One of the limitations of multiplex IHC is the fixed number of antibodies which can be used to query tissue in-situ. In order to maximize coverage of both lymphoid and myeloid cells, we chose an antibody to CD163, a scavenger receptor found on both macrophages and dendritic cells. We were unable to separate the two populations in our analysis but do provide additional data looking at engagement of CTLs and APCs in the HLA-DR high subset. Interestingly, there was a significantly greater degree of engagement, highlighting the enhanced pro-inflammatory environment in these tumors. The text and figures have been updated to reflect this additional data. 

  • Optional: The authors should also study the IL-10 and TGF-beta expression levels in HLA-DR high and low samples.

While we agree this would be very interesting to further study, we are limited in the access to tissue in these patients which prohibits us from this more in-depth analysis.

Minor Points:

  • The authors did not mention the HLA-DR antibody used for the study.

This has now been included in the supplemental table of antibodies

  • Inconsistent use of MHC-II & MHC2 and Figure & figure

This has been corrected

Reviewer 2 Report

In contrast to MHC class 1, our knowledge related to MHC class 2 in the tumor microenvironment is very poor. The major goal of the present article is to start to fill this gap, by studying and describing the expression of HLA-DR antigens and its association with certain biological characteristics of the tumor microenvironment.

The study is well designed, the appropriate number of patients (and thus, tumor samples) is recognized and welcome by the reviewer - studies with similar methodology are often underpowered. The focus is clear, the methods are adequate and well-described, the results are clearly presented, the figures, photos, graphs are illustrative. The applied statistical tests are also correct.

The study sheds light on some aspects of HLA-DR expression and immune cell infiltration around the tumor, or distance between different cells in the tumor microenvironment. However, interpretation of certain results (and their potential application in enhancing the efficacy of tumor therapy) requires further research on this field. The discussion thoroughly and appropriately puts the results into context, and emphasizes the mentioned need for further research.

Altogether this is a well-designed and performed study on a topic with a potential impact on improving tumors therapies in the future. Hopefully, as a consequence of this research, the area receives increased attention, and more and more publications will be produced on this topic.

Author Response

We thank the reviewer for their thoughtful analysis of our study.

Round 2

Reviewer 1 Report

The authors addressed all the concerns raised by the reviewer.